# Comprehensive Assessment of the Time Course of Biomechanical, Electrophysiological and Neuro-Motor Effects after Botulinum Toxin Injections in Elbow Flexors of Chronic Stroke Survivors with Spastic Hemiplegia: A Cross Sectional Observation Study

**DOI:** 10.3390/toxins14020104

**Published:** 2022-01-28

**Authors:** Yen-Ting Chen, Yang Liu, Chuan Zhang, Elaine Magat, Ping Zhou, Yingchun Zhang, Sheng Li

**Affiliations:** 1Department of Physical Medicine and Rehabilitation, University of Texas Health Science Center at Houston, Houston, TX 77030, USA; cheny@nsuok.edu (Y.-T.C.); elaine.m.magat@uth.tmc.edu (E.M.); 2TIRR Memorial Hermann Hospital, Houston, TX 77030, USA; 3Department of Health and Kinesiology, Northeastern State University, Broken Arrow, OK 74014, USA; 4Department of Biomedical Engineering, University of Houston, Houston, TX 77204, USA; yliu77@uh.edu (Y.L.); czhang19@uh.edu (C.Z.); yzhang94@Central.UH.EDU (Y.Z.); 5Faculty of Biomedical and Rehabilitation Engineering, University of Health and Rehabilitation Sciences, Qingdao 266024, China; dr.ping.zhou@outlook.com

**Keywords:** Botulinum toxin, stroke, spasticity, motor control, stretch reflex

## Abstract

Botulinum neurotoxin (BoNT) is commonly used to manage focal spasticity in stroke survivors. This study aimed to a perform comprehensive assessment of the effects of BoNT injection. Twelve stroke subjects with spastic hemiplegia (age: 52.0 ± 10.1 year; 5 females) received 100 units of BoNT to the spastic biceps brachii muscles. Clinical, biomechanical, electrophysiological, and neuro-motor assessments were performed one week (wk) before (pre-injection), 3 weeks (wks) after, and 3 months (mons) after BoNT injection. BoNT injection significantly reduced spasticity, muscle strength, reflex torque, and compound muscle action potential (CMAP) amplitude of spastic elbow flexors (all *p* < 0.05) during the 3-wks visit, and these values return to the pre-injection level during the 3-mons visit. Furthermore, the degree of reflex torque change was negatively correlated to the amount of non-reflex component of elbow flexor resistance torque. However, voluntary force control and non-reflex resistance torque remained unchanged throughout. Our results revealed parallel changes in clinical, neurophysiological and biomechanical assessment after BoNT injection; BoNT injection would be more effective if hypertonia was mainly mediated by underlying neural mechanisms. BoNT did not affect voluntary force control of spastic muscles.

## 1. Introduction

Spasticity and weakness are disabling motor impairments after stroke. In contrast to weakness that occurs at the stroke onset, spasticity emerges in the subacute stage. The mean time to detect spasticity is 34 days after stroke onset [1]. Spasticity continues to develop and worsen over time. The prevalence of spasticity is about 19.0% by 3 mons [2], and 43.2% by 12 mons after stroke [3]. In the chronic stage, spasticity is present in up to 97% of stroke survivors with moderate to severe motor impairments [4]. Emergence and development of post-stroke spasticity reflects a maladaptive plastic process. Secondary to damage to the motor cortex and/or to the corticobulbar pathways after stroke, excitability of the contralateral medial cortico-reticulo-spinal pathways becomes unopposed and gradually upregulated, thus resulting in un-opposed excitatory inputs to spinal reflex circuits [5,6,7]. In the presence of externally imposed muscle stretch, the phenomenon of velocity-dependent stretch reflex responses, i.e., spasticity, is manifested [8,9]. On the other hand, this pathophysiological process of hyperexcitable descending brainstem pathways is shared by other motor impairments, e.g., inappropriate muscle activation [7,9]. During sustained voluntary elbow flexion on the spastic-paretic side of stroke survivors, Rymer and colleagues have reported that spinal motor neurons are hyperexcitable and even spontaneously firing. Spontaneous firing increases with the level of exertion [10,11], and continues after voluntary contraction ends [12]. The spasticity-related spontaneous motor neuron firing is not under voluntary control, and could be viewed as “motor noise” during voluntary activation. As such, they may make it difficult for stroke survivors to perform voluntary actions with spastic muscles, such as to initiate or terminate a hand grip [13,14] or maintain a constant force output [12,15,16,17]. In other words, spasticity interferes with motor performance of spastic muscles [9].

Botulinum toxin (BoNT) therapy is the preferred treatment option for focal spasticity after a stroke [18]. BoNT blocks pre-synaptic release of acetylcholine at the neuromuscular junction, thus leading to muscle relaxation and spasticity reduction [19,20,21]. BoNT therapy has an established profile of clinical effects. The effect starts several days after an injection, reaches its peak around 3–4 wks, and lasts about 3 mons [22,23]. Therefore, patients usually require a repeat BoNT injection every 3–4 mons to maintain the effect [18,24]. A recent meta-analysis of 40 clinical trials over 30 years has demonstrated robust results of spasticity reduction effects after BoNT injections [25].

Different methods have been used to quantify the BoNT effects on spasticity reduction. Clinical scales, such as modified Ashworth scale (MAS) and the Tardieu scale, are commonly used in clinical trials. However, these clinical scales do not differentiate neural, i.e., stretch reflex-mediated, and non-neural, i.e., stiffness from muscles and connective tissues, components of spastic hypertonia. In the laboratory setting, quantitative analysis of resistance to motorized stretch is able to assess contributions of different components to the overall spastic hypertonia [26,27,28,29]. In this approach, reflex torque is defined as the difference between the peak resistance torque during fast stretching (e.g., 100°/s) and during slow stretching (e.g., 5°/s), reflecting the stretch reflex-mediated neural component, while the resistant torque during slow stretching is considered as the non-neural property of the spastic muscles. This differentiation is important because BoNT injections also induce muscular changes and alter muscle stiffness [30]. It was found that both the MAS score and reflex torque of elbow flexor muscles significantly decreased 3 wks after BoNT injections, the non-reflex torque remained unchanged [17]. In a longitudinal study, the reflex torque of wrist flexors of stroke survivors decreased at 4 wks after BoNT injections, and returned to baseline levels at 12 wks. Similarly, the non-reflex torque remains unchanged after the treatment [17]. Electrophysiological assessment has also been used to quantify the BoNT effects [31,32,33]. Zhang et al. [33] measured compound muscle action potentials (CMAP) from the biceps muscles before, 3 wks and 3 mons after BoNT injections to the biceps muscles. The authors reported both the MAS scores and CMAP values decreased at 3 wks and returned to baseline levels at 3 mons after the injections.

Muscle weakness after BoNT injections is expected because of the neurotransmitter blockade effects of BoNT. The effects of BoNT injections on motor performance of spastic muscles remain controversial. A recent meta-analysis revealed robust evidence of a lack of meaningful functional improvement in arm and hand use after BoNT injections [25]. A few studies have reported improved motor function after BoNT injections in a subgroup of stroke survivors [12,33,34]. Moreover, our recent study has demonstrated that stroke survivors were able to maintain the steady force output during voluntary elbow flexion, i.e., unchanged force variability, despite the fact that elbow flexors became weaker at 3 wks after BoNT injections [17].

There are only few laboratory studies that characterized the time course of different BoNT effects, e.g., on voluntary activation capacity [35], difference in reflex and voluntary responses [32], neural vs. non-neural response [34] and electrophysiological response [33]. As described above, BoNT injections have multiple effects on the spastic muscles, including neural and non-neural components of hypertonia, muscle strength, and motor performance. Accordingly, we aimed to perform a comprehensive assessment of the time course of biomechanical (reflex and baseline torque), neurophysiological (CMAP) and neuro-motor (force control) effects after BoNT injections in the elbow flexors of chronic stroke survivors with spastic hemiplegia. Specifically, we hypothesized that spasticity reduction is accompanied by parallel changes in reflex torque and CMAP after BoNT injections, but voluntary force control remains unchanged.

## 2. Results

### 2.1. Clinical Assessment of BoNT Effects

Demographic and other information of all patients were summarized in Table 1. Twelve stroke subjects completed all three visits (pre-injection, 3-wks, and 3-mons).

#### 2.1.1. Elbow Flexor Spasticity

There was a significant main effect of TIME for the MAS scores in the Friedman test (*p* < 0.01). Specifically, BoNT injection significantly reduced MAS scores during the 3-wks follow-up compared to pre-injection visit (*p* < 0.01). Furthermore, the MAS scores significantly increased during the 3-mons follow-up compared to the 3-wks follow-up (*p* = 0.03). Notably, The MAS scores were significantly greater during the pre-injection visit compared to the 3-mons follow-up visit (*p* = 0.05). In other words, the MAS scores at 3 mons after BoNT injections did not fully return to the baseline.

#### 2.1.2. Muscle Strength—Maximum Voluntary Contraction (MVC) Tasks

There was significant main effect of TIME on the MVC force of spastic elbow flexors (*p* < 0.01, partial η^2^ = 0.62; Figure 1). Specifically, The MVC force was significantly lower during the 3-wks follow-up (9.98 ± 4.64 N-m) compared to pre-injection (12.73 ± 6.37 N-m; *p* < 0.05) and 3-mons follow-up visits (13.37 ± 4.23 N-m; *p* = 0.02). On the other hand, one-way repeated measure ANOVA showed that there was no significant change of the elbow flexion MVC force on the contralateral side among three visits (pre-injection: 38.13 ± 12.88 N-m, 3-wks: 38.65 ± 12.24 N-m, 3-mons: 39.07 ± 11.5 N-m; *p* > 0.05).

### 2.2. Biomechanical Assessment of BoNT Effects

The resistance torque during 100°/s passive stretch from all three visits of one representative patient was demonstrated in Figure 2a. There was a significant main effect of TIME in one-way ANOVAs with repeated measure for both total torque (*p* = 0.02; partial η^2^ = 0.39; Figure 2b) and reflex torque (*p* = 0.03; η^2^ = 0.35; Figure 2c) but not baseline torque (*p* = 0.39, partial η^2^ = 0.07; Figure 2d). Post-hoc analyses revealed that total torque was significantly lower during the 3-wks follow-up (6.18 ± 2.79 N-m) compared to the pre-injection visit (7.73 ± 3.17 N-m; *p* < 0.01) and 3-mons follow-up (7.39 ± 3.52 N-m; *p* = 0.02). Furthermore, reflex torque was also significantly lower during the 3-wks follow-up (2.26 ± 1.47 N-m) compared to the pre-injection visit (3.48 ± 2.05 N-m; *p* < 0.01). However, the difference between the 3-wks follow-up and 3-mons follow-up (3.14 ± 2.04 N-m) was only close to being statistically significant (*p* = 0.052). Moreover, the reflex torque change between pre-injection and 3-wks follow-up negatively correlated with the percentage of baseline torque to the total torque during the pre-injection visit (r = −0.74, *p* < 0.01; Figure 3). This result indicated that the less mechanical component (i.e., baseline torque) of spasticity over the total spasticity, the better improvement (more reduction of reflex torque) after BoNT injection.

### 2.3. The Effect of BoNT on Neuro-Motor Performance

Three-way repeated measure ANOVA tests with factors of LIMB, force level and TIME were performed for force variability (CV of force). There was a significant main effect of LIMB (*p* = 0.01, partial η^2^ = 0.58) and significant interactions between LIMB and force level (*p* = 0.03, partial η^2^ = 0.47) (Figure 4). Post-hoc analyses revealed that the significant difference in elbow flexion force control between the impaired and contralateral side was only during 10% of MVC tasks (impaired side: 3.28 ± 3.13%, contralateral side: 1.35 ± 0.96%; *p* < 0.05). There were no other significant main effects or significant interactions. CV remained unchanged at the 3-wks and 3-mons visits.

### 2.4. Neurophysiological Assessment of BoNT Effects—CMAP Amplitude

There was significant main effect of TIME in one-way ANOVA analysis with repeated measure for CMAP amplitude (*p* < 0.01, partial η^2^ = 0.49; Figure 5). Specifically, post-hoc analyses revealed that CMAP amplitude was significantly lower during the 3-wks visit (6.61 ± 3.13 mV) compared to the pre-injection visit (8.97 ± 4.08 N-m; *p* < 0.01) and 3-mons follow-up visit (8.13 ± 3.60 N-m; *p* = 0.03). Furthermore, there was no significant difference between percent change of CMAP amplitude (−21.42 ± 21.81%) and percent change of reflex torque (−34.01 ± 29.60%) during the 3-wks visit.

## 3. Discussion

In this study, a cohort of 12 chronic stroke survivors with elbow flexor spasticity received botulinum toxin (BoNT) to spastic biceps muscles. A comprehensive assessment of BoNT effects was performed before, at 3-wks, and at 3-mons after injections, including clinical assessment (MAS and elbow flexor strength), biomechanical and neurophysiological assessment, and neuro-motor performance assessment. In this first comprehensive assessment study of the time course of BoNT effects, we observed parallel changes in assessments longitudinally. At about 3 wks after injections, BoNT therapy resulted in decreased muscle strength, spasticity reduction in the MAS score, decreased total torque and decreased reflex torque, as well as decreased CMAP amplitude. These values returned towards the baseline levels at 3 mons after injections. In general, these results from laboratory biomechanical and neurophysiological assessments confirmed the established clinical profile of BoNT effects (see Introduction for details). In addition to these expected outcomes, we also observed that there were no changes in the baseline torque, and force variability remained unchanged throughout longitudinal assessments. These novel findings provide new insights into our understanding of BoNT effects and their potential clinical implications.

### 3.1. BoNT Effects on Elbow Flexor Spastic Hypertonia

As mentioned in the Introduction, spastic hypertonia has different components. In this study, BoNT injections had different effects on neural and non-neural components of elbow flexor spastic hypertonia. The reflex torque significantly decreased during the 3-wks visit (2.26 ± 1.47 N-m) from the pre-injection level (3.48 ± 2.05 N-m; *p* < 0.01). These results confirmed our recent finding that BoNT injections significantly decreased the reflex torque during the 3-wks visit [17]. Furthermore, the finding of a similar change in the reflex torque and CMAP amplitude indicates that reduction in the reflex torque is attributable to the denervation effect of BoNT injections, i.e., the neural component of spastic hypertonia. However, the reflex torque did increase, but not fully return during the 3-mons visit (3.14 ± 2.04 N-m, *p* = 0.052). In contrast, the baseline torque remained unchanged during the 3-wks and 3-mons visits after BoNT injections, while the total torque significantly decreased during the 3-wks visit, and returned during the 3-mons visit. The disparity in the total torque, reflex torque and baseline torque during the 3-mons visit may suggest that the reflex torque via stretch reflex circuits may take a longer time to recover, even though the CMAP amplitude has returned to the pre-injection level. It has been reported that the reflex responses take a longer recovery time [32]. A longer period of longitudinal investigation needs to be used in the future studies.

The finding of unchanged baseline torque after BoNT injections in not trivial. The baseline torque reflects the non-neural component of spastic hypertonia. The passive stiffness includes viscoelasticity of the muscle [36,37], changes in muscle fiber property [38], and accumulation of extracellular deposits [39]. BoNT injections cause chemodenervation effects on the injected spastic muscles via presynaptic blockade of neurotransmitters at the neuromuscular junctions [19,20,21]. Animal studies have shown adaptive changes in the muscles after BoNT injections, including increased collagen content and passive resistance [40,41], muscle fiber loss and atrophy [30]. Collectively, these changes contribute to increased passive stiffness after BoNT. Gross muscle morphology takes more than 12 mons to recover [42]. Similar findings were observed in human subjects. In a recent study that compared passive properties of spastic wrist and finger flexors in chronic stroke survivors, no significant differences in passive torques between affected and contralateral limbs were found in those who never received BoNT. In contrast, passive torques were significantly higher and passive range of motion of wrist and finger extension was significantly smaller in the affected limb than the contralateral limb in those who received BoNT injections to these muscles, on average, 4 and 1/2 years ago [43]. In another study [44], stroke subjects received either BoNT or placebo (normal saline) injections when spasticity was first detected in the arm flexors. The mean time to injections was 18 days after stroke. The authors reported that elbow flexor spasticity was lower, the range of motion was higher (i.e., less passive resistance and contracture) at 2 to 12 wks after injection in the BoNT group as compared to the placebo group. In this early BoNT-injection study [44], therapeutic modalities, such as splinting, were used in this study. The result of unchanged baseline passive torque over three mons after BoNT injection in our study suggests that the adaptive changes in the injected muscles take time to develop in chronic stroke survivors.

### 3.2. Effects on Neuro-Motor Control of Spastic Muscles

Production and maintenance of a steady force is essential in performing motor tasks in activities of daily living, e.g., holding a cup of coffee. Force variability is a quantitative measure of such neuro-motor performance. Force variability is primarily dependent on the strength of a muscle in healthy subjects—the stronger the muscle, the less force variability is observed [45]. It is well-established that force variability was greater on the spastic paretic side than on the contralateral side of stroke survivors [12,15,16,17]. This is mainly attributed to the fact that spastic muscles are weaker than the muscles on the contralateral side. In a recent study [17], we reported that force variability of voluntary elbow flexion remained unchanged 3 wks after BoNT injections to the biceps muscles. It was inferred that, although spastic elbow flexors became weaker after BoNT injections, BoNT also minimized spasticity-related spontaneous firing of motor neurons, thus “motor noise”. As a result, unchanged force variability was observed after BoNT injection. In this study, we confirmed and expanded that force variability remained unchanged during the 3-mons follow up visit when the BoNT effect is worn off. The finding is significant in that neuro-motor control of spastic muscles remains unchanged after BoNT injections, regardless of its effect on muscle strength. Its clinical implications are discussed next.

### 3.3. Clinical Implications

Over decades of clinical application of BoNT therapy, numerous studies have provided robust evidence that BoNT therapy is effective in spasticity reduction [25]. Based on a comprehensive assessment throughout the time course after BoNT injections, this study provides biomechanical and neurophysiological evidence to support the established clinical profile of BoNT effects on spasticity reduction in an injection cycle. Furthermore, results from this study also shed light on how to maximize the BoNT effects.

One strategy is to combine BoNT injection with therapeutic physical modalities [46]. Physical modalities have been shown to be effective in reducing hypertonia and increasing range of motion, such as splinting and casting [47,48,49,50,51]. However, a systematic review on the use of upper extremity casting found high variability in casting protocols which indicates no consensus in technique [52]. Our results showed the reflex torque decreased at 3 wks after BoNT injection, while the baseline passive torque remained the same. It is thus advisable that casting is best performed before the BoNT effect reaches its peak when the total resistance is the lowest, as such, casting-induced prolonged stretching of spastic muscles can delay the return of reflex torque, thus maximizing the BoNT effect [53].

Unaltered neuro-motor control of spastic muscles after BoNT injections is also a clinically meaningful finding. The term “therapeutic weakness” best explains its implication [54]. In a selected group of stroke survivors with the ability to voluntarily activate spastic muscles, BoNT injections could effectively improve motor function, such as improved force initiation and relaxation [13,55]. For example, stroke survivors with spastic finger flexors become weaker after injection, but are able to open and close the hand more efficiently [13].

One major limitation of the present study is sample size. In order to control the severity of spasticity, the dosage of BoNT injection, and injection location, there were only few percentages of the patients qualified for our study. For each of ANOVA analyses with repeated measure, the partial η^2^ value was calculated. The partial η^2^ values from both significant and non-significant findings indicated that we have medium to large effects. Furthermore, the present study only focused on one muscle (biceps), so the results may not be directly generalized to other muscle groups. Another limitation is the measurement of MAS. Assessment of MAS is subjective, and it was not double-blinded in this study, so the results of MAS changes may be influenced by the examinee’s judgment. In this study, spasticity was assessed by reflex torque using a quantitative biomechanical method. Parallel changes in reflex torque and MAS after BoNT injection indicated that the MAS findings in this study are likely a reflection of true changes after BoNT injection. However, a longer duration of follow up is needed to better reflect changes after BoNT injections.

## 4. Conclusions

Our results provide evidence of decreased muscle strength, spasticity reduction in the MAS score, and decreased total torque and decreased reflex torque, as well as decreased CMAP amplitude at 3 wks after BoNT injections to the biceps muscles. These values returned towards the baseline levels at 3 mons after injections. BoNT injections did not alter the baseline passive torque or force variability of voluntary elbow flexion throughout the time course. Future studies in stroke survivors with a wide age range and different fitness status, and multiple muscle groups and larger sample size are needed to expand our knowledge regarding the effects of BoNT injection on spasticity and motor performance.

## 5. Methods

### 5.1. Participants

Twelve stroke survivors with spastic hemiplegia participated in this study. Inclusion criteria included (1) hemiplegia secondary to an ischemic or hemorrhagic stroke; (2) at least six mons post-stroke; (3) rated as Modified Ashworth Scale (MAS) score 2 or 3 during the baseline assessment; (4) elbow flexor spasticity requiring 100 units of onabotuliumtoxinA or incobotulinumtoxinA injections to the spastic biceps brachii muscle; and (5) able to understand and follow instructions related to the experiment. Exclusion criteria included stroke survivors with: (1) more than one stroke; (2) increased resistance from other etiologies, such as rigidity and contracture; (3) a history of prior musculoskeletal injury involving upper extremity; and (4) other comorbidities that may cause spasticity, such as spinal cord injury. The procedure of this study was approved by the Committee for the Protection of Human Subjects at the University of Texas Health Science Center at Houston and the University of Houston. A total of 504 patients were screened between November 2017 and November 2019 by the physicians in TIRR Memorial Hermann Hospital. Fifteen stroke patients qualified and agreed to participate in the study. All participants provided written informed consent before participating in the study. During the study period, participants who had any changes of medications and/or treatment which would affect spasticity or motor performance would be excluded. Two subjects withdrew from the study voluntarily, and one subject was excluded due to the change of medications during the experiment period that affect muscle functions. One subject was naïve to the BoNT injection, while 11 out of 12 subjects received BoNT injections at least 2 times with satisfactory responses prior to this study.

### 5.2. Experimental Set-Up

Subjects sat comfortably on a height-adjustable chair. The tested arm was secured in a customized device in the following position: the shoulder joint was placed approximately in 45° of flexion and 30° of abduction. The wrist and fingers were in the neutral position. The elbow joint during MVC and motor performance control tasks was fixed at 90° of elbow flexion, and was varied during passive stretch tasks as described in the experimental protocol session. A 20-inch computer monitor (Model: 2001FP, Dell Computer Corp., Round Rock, TX, USA) was set approximately one meter in front at the subject’s eye level for displaying visual feedback during motor performance tasks.

Force signals (MVC, motor performance, and passive stretch) were measured with a torque sensor (Model: TRS-500, Transducer Techniques, Temecula, CA, USA). A servo motor (Model: FHA-25C-50-US250) was used to passively stretch elbow flexor muscles. The axis of rotation of the elbow joint was aligned with the axis of rotation of both torque sensor and servo motor. Custom-written programs in LabView^®^ (National Instrument™ Inc., Austin, TX, USA) were used to for collecting force signals during MVC, motor performance, and passive stretch tasks.

The CMAP of the biceps branchii was evoked by transcutaneous stimulation of the musculocutaneous nerve via a DS7A (Digitimer Ltd., Hertfordshire, UK). Electrical stimulation was triggered manually by square pulses with pulse width of 200 μs. The optimal stimulation spot was determined by observing the amplitude and morphology of elicited CMAP when stimulating the nerve between the coracobrachialis muscle and the short head of biceps muscle with low to moderate stimulation intensity. The stimulation electrode remained at this determined location and increased the stimulation intensity with 10 mA step until the supramaximal response was acquired. The constant room temperature at TIRR Memorial Hermann Hospital ensured that the surface skin temperature was above 32 °C.

The force signals were sampled at 1000 Hz with an NI-DAQ card Model: PCI-6229, National Instruments, Austin, TX, USA). The EMG signals were amplified with a gain of 26.55, and sampled at 2048 Hz and recorded through a 136 channel Refa amplifier (TMSi, Enschede, The Netherlands).

### 5.3. Experimental Protocol

There were three experimental sessions. Within one wk before the scheduled BoNT injection, the participants were scheduled for the first session (pre-injection). The second session was scheduled 3–4 wks after the injection (3-wks follow-up), and the third session was scheduled 3 mons after the injection (3-mons follow-up). During each session, the following assessments were performed: (1) Modified Ashworth Scale (MAS) of spastic elbow flexors; (2) maximum voluntary contraction (MVC) of elbow flexors on each side; (3) force control performance of elbow flexion on each side; (4) passive stretch torque responses on the affected side; and (5) amplitude of the compound muscle action potentials (CMAP) on the affected side.

MAS of spastic elbow flexors was assessed by the same researcher during three visits. The participants were asked to perform unilateral elbow flexion MVC tasks on both sides. Subjects were asked to reach and hold the maximum force for 3–5 s. Three trials were performed for each side. The highest value among three attempts was determined as the MVC force. At least 1-min rest was provided between consecutive MVC attempts.

After MVC tasks, each subject performed unilateral isometric elbow flexion tasks at 10%, 30%, and 50% of their MVC for both sides. A target horizontal red line was displayed on the monitor in the beginning of each trial. During each 12-s trial, the force trace shown as a white line started running from the left to right on the monitor. Subjects were asked to match the white trace (their force) to the red line (target) as precisely as they can. Three trials were performed for each side under each force level.

During biomechanical assessment, the impaired elbow joint was passively stretched using an established experimental paradigm [17,56]. A customized device was used to passively extend the forearm for 50° and then move it back to the initial position. During the pre-injection visit, the angle was recorded when a catch occurred during a quick manual passive stretch. The initial position was determined as this catch angle minus 10 degrees (i.e., more flexed). The initial position was kept the same for all 3 visits. Both slow (5°/s) and fast (100°/s) stretch speeds were used. Subjects were asked to relax during the passive stretch tasks and at least 1-min rest was given between trials. For each stretch speed, three trials were collected.

CMAPs were obtained after passive stretch tasks. The supramaximal CMAP amplitude was confirmed when the increase of the electric stimulation could not increase the CMAP amplitude, as used in our previous studies [33,57].

### 5.4. Data Analysis

Data were analyzed using custom-written Matlab programs (MathWorks, Natick, MA, USA). The raw torque signal was low-pass filtered at 10 Hz with a fourth-order, zero-lag Butterworth digital. The EMG signals were band-pass filtered (10–500 Hz) and notch filtered (stopband 59.5–60.5 Hz). The following parameters were calculated:

#### 5.4.1. Neuro-Motor Performance

Neuro-motor performance was quantified by the coefficient of variation (CV) of the elbow flexor isometric force produced by the subjects. CV of force was defined as the standard deviation of the force divided by the mean of the force. The middle 2 s of each trial was used to calculate the performance. A larger value of CV indicates higher fluctuation of the force and worse performance [17]. The average value of all three trials for each force level was used to indicate the force control performance for both impaired and non-impaired sides.

#### 5.4.2. Resistance Torque

The calculation of the resistance torque during a passive stretch task was described in our previous study [17]. The resistance torque during 5°/s passive stretch (slow stretch) was considered as the non-reflex property of the spastic muscle (baseline torque), while the resistance torque during 100°/s passive stretch (fast stretch) was considered as the overall hypertonia that contains both non-reflex and reflex properties of spastic muscles (total torque). The difference between the total torque and baseline torque was defined as the reflex torque.

#### 5.4.3. CMAP

Before further analyses, the direct current component was removed and the stimulation artifact was suppressed in the EMG signals based on a previously reported protocol [58]. The monopolar CMAP amplitude was defined as the difference between the baseline amplitude and the negative peak value elicited by an electrical stimulus. The largest CMAP amplitude among the 128 sEMG channels was characterized to represent the CMAP amplitude of a subject during a visit.

### 5.5. Statistical Analysis

The following dependent variables were used: (1) MAS; (2) MVC force; (3) CV; (4) baseline torque; (5) total torque; (6) reflex torque; and (7) CMAP amplitude. MVC force and motor control performance of the impaired side were not collected from two patients due to no detectable elbow flexion force. MVC and CV on the contralateral side were not collected from one subject due to time restriction. Non-parametric Friedman test was used to test the main effect of TIME (pre-injection, 3-wks follow-up, and 3-mons follow-up) for MAS scores. One-way repeated measure ANOVA with the factor of TIME was used to test the effect of BoNT injection on MVC force, baseline torque, total torque, reflex torque, and CMAP amplitude on the impaired side. For statistical analysis purposes, we used 0, 1, 2, 3, 4, 5 to represent a MAS value of 0, 1, 1+, 2, 3, 4, respectively. Three-way repeated measure ANOVA with the factors of TIME, LIMBS (impaired and non-impaired), and FORCELEVEL (10%, 30%, and 50%) was used to test the effect of injection on motor control performance. Fisher’s least significant difference (LSD) post-hoc analyses were performed for any significant main effect or interaction. Analyses were performed with the IBM SPSS Statistics 25 (IBM Corp., Armonk, NY, USA). The alpha level for all statistical tests was 0.05. Data were reported as mean ± standard deviation (SD) within the text and as mean ± standard error mean (SEM) in the figures.

## Figures and Tables

**Figure 1 toxins-14-00104-f001:**
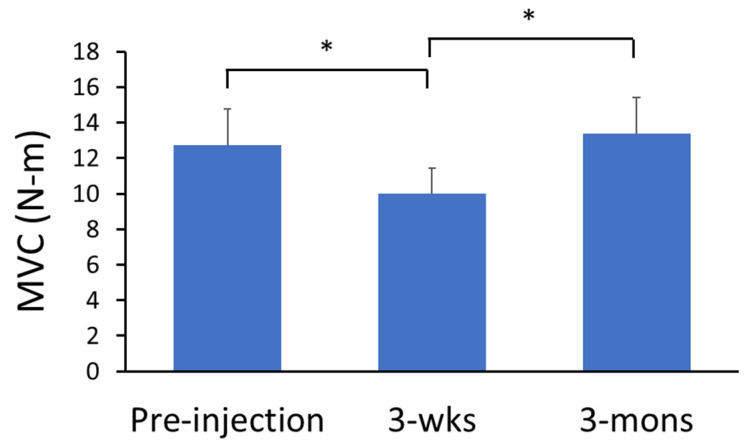
MVC of spastic elbow flexors. * Indicates statistically significant difference (*p* < 0.05).

**Figure 2 toxins-14-00104-f002:**
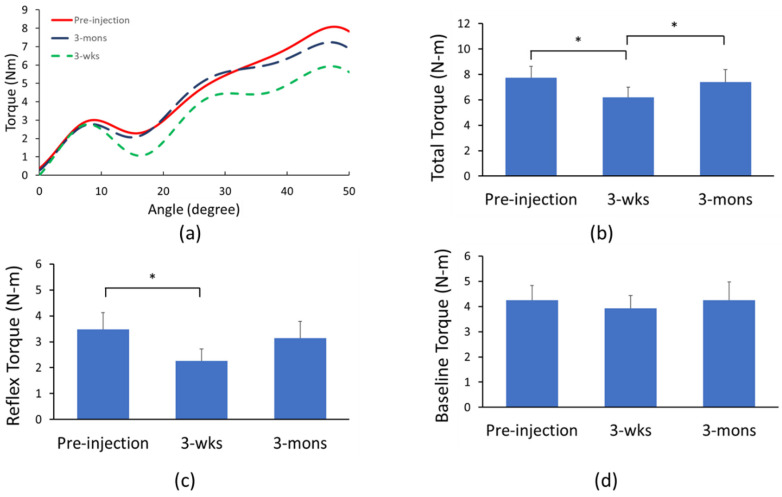
Passive stretch torque before and after BoNT injection. (**a**) Representative trials of total torque during three visits; (**b**) total torque; (**c**) reflex torque; and (**d**) baseline torque. * Indicates statistically significant difference (*p* < 0.05).

**Figure 3 toxins-14-00104-f003:**
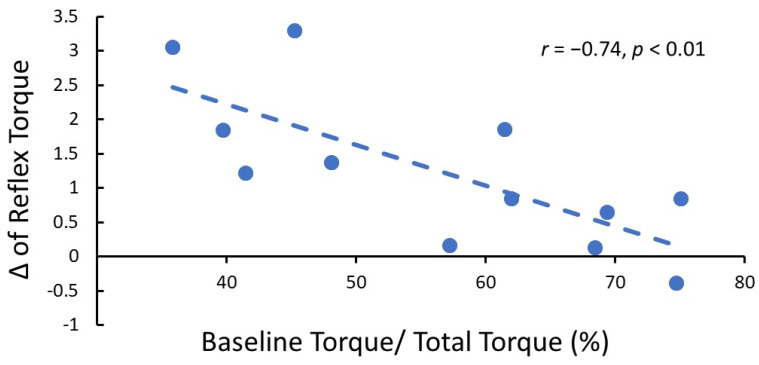
Correlation between change of reflex torque after BoNT injection and the ratio of baseline torque to total torque (r = −0.74, *p* < 0.01) during the 3-wks visit. This result indicated that the spastic muscle with less non-reflex property would benefit from more reduction in reflex torque.

**Figure 4 toxins-14-00104-f004:**
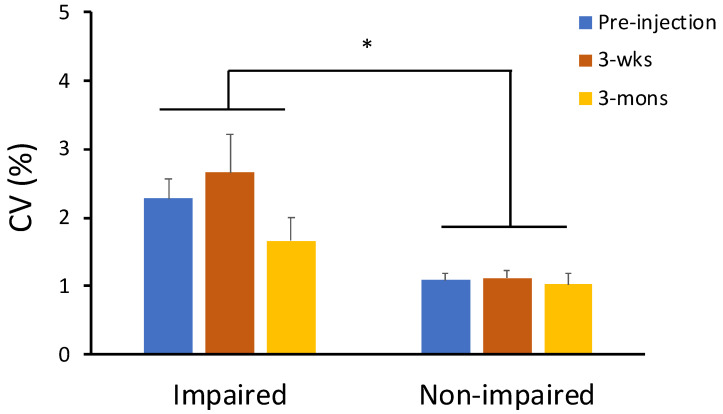
The effect of BoNT injection on force variability (CV). On average, CV was significantly greater on the impaired side than on the non-impaired side. However, CV remained statistically unchanged at 3 wks and 3 mons after BoNT injections. * Indicates statistically significant difference (*p* < 0.05).

**Figure 5 toxins-14-00104-f005:**
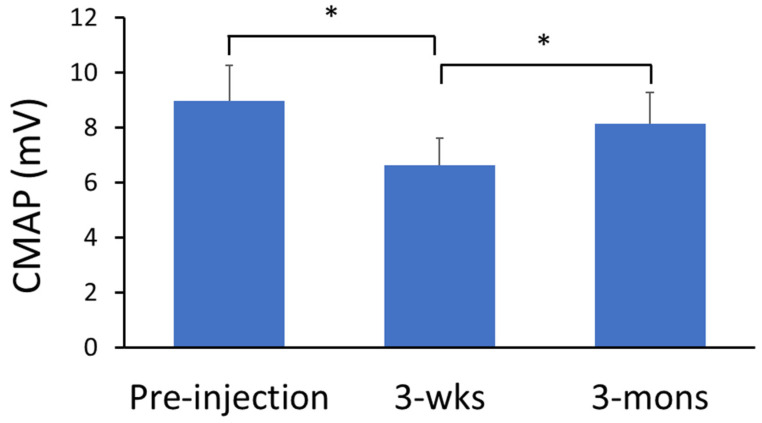
The effect of BoNT injection on CMAP. CMAP amplitude reduced during the 3-wks visit compared to the pre-injection visit. During the 3-mons visit, the CMAP amplitude increased significantly compared to the 3-wks visit and had no difference to the CMAP amplitude before BoNT injection (pre-injection). * Indicates statistically significant difference (*p* < 0.05).

**Table 1 toxins-14-00104-t001:** Characteristics of the participated patients. MAS: modified Ashworth scale.

ID	Age	Gender	History of Stroke (Mons)	Paretic Side	Dominant Side	Elbow Flexor MAS (Pre-Injection)	Elbow Flexor MAS (3-Wks)	Elbow Flexor MAS (3-Mons)	Lesion Type
1	60	M	23	Left	Right	2	2	2	Hemorrhagic
2	52	F	104	Left	Right	2	1+	1+	Ischemic
3	40	F	77	Right	Right	2	1+	2	Ischemic
4	53	M	68	Left	Right	2	1+	2	Hemorrhagic
5	49	M	89	Left	Right	2	2	2	Hemorrhagic
6	63	M	137	Left	Right	2	1+	1+	Ischemic
7	49	M	7	Right	Right	2	1+	2	Ischemic
8	40	M	39	Right	Right	2	1+	1+	Hemorrhagic
9	68	F	7	Left	Right	2	2	2	Ischemic
10	65	M	29	Left	Right	2	1+	2	Hemorrhagic
11	39	F	76	Right	Right	2	1	2	Hemorrhagic
12	46	F	117	Right	Right	3	2	2	Ischemic

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
