# Peer review of "Comprehensive Assessment of the Time Course of Biomechanical, Electrophysiological and Neuro-Motor Effects after Botulinum Toxin Injections in Elbow Flexors of Chronic Stroke Survivors with Spastic Hemiplegia: A Cross Sectional Observation Study"

_toxins, 2022, doi:10.3390/toxins14020104_

Round 1
Reviewer 1 Report
This is an interesting paper.
This will give more insights to the reader regarding neuromotor effects of the botulinum toxin.
Author Response
We thank the Reviewer for encouragement and support.
We have gone through the manuscript carefully to improve the readability, see numerous changes in the revised manuscript.
Reviewer 2 Report
An excellent work,well presented and explained, only one suggestion: the median age of the patients is low, and there is not included the fitness status. I have the immpression that older patients or different fitness status would alterate the results.so maybe this option should be refered (lines 289-291)
Author Response
We thank the Reviewer for this positive comment. We have changed the statement (lines 289-291) to reflect the Reviewer’s suggestions.
Reviewer 3 Report
The second submission is improved, including a new paragraph on the study's weaknesses. This paragraph is important for readers, especially those unfamiliar with this kind of study. In my opinion, the manuscript was suitable for publication in the first version, and now I repeat my suggestion.
Author Response
Thank you for your support
Reviewer 4 Report
In this study, a cohort of 12 chronic stroke survivors with elbow flexor spasticity received botulinum toxin (BoNT) to spastic biceps muscles. A comprehensive assessment of BoNT effects was performed before, at 3-wks, and at 3- months after injections, including clinical assessment (MAS and elbow flexor strength), biomechanical and neurophysiological assessment, and neuro-motor performance assessment. BoNT did not affect voluntary force control of spastic muscles
The work is carefully and well done,
The study is well done, the data has been properly analyzed. This is a study, is suitable for publication in Antioxidants.
Introduction section: The authors substantiated the design of this interventional study by referring to the appropriate references.
Materials and methods: The study was well designed. The methods were properly described.
The results are clearly presented and correctly analyzed. The discussion based on well-chosen literature. Conclusions are supported by the results.
I have also a few minor comments:
Small typographical errors
All the abbreviations used in text should be presented at the begining in form of the list to make it easier for readers.
It could be clearer to present it in the form of table or diagram with design
Author Response
Thank you for your support.
The typos have been corrected throughout.
A list of abbreviation is added.
We felt that the description was sufficient, since it was a pre-, post- and followup design.
Reviewer 5 Report
This is a well-written manuscript that is clear and which provides new data on the use of BoNT for spasticity treatment in a carefully selected group of patients.
My minor comments are as follows.
The abbreviation "mons" should be changed to months.
Figures 4 & 5 are not correctly mentioned in the main text.
Lines 180-181 & 286-287 These state about a return to baseline levels at 3 months, including MAS. But lines 111-112 state that MAS scores did not return to baseline after 3 months.
Lines 209-211 Not required - repeats earlier
Line 215 Citation 44 relates to a rat study, not a human study. So this statement about translation to humans needs detailed justification. How are such animal studies translational to humans? Evidence indicates much higher doses of BoNT are need for animal effects than for humans.
Lines 215-218 Citation needed
Lines 250-254 The present study only examined 2 time points after BoNT injections. Therefore these statements are a little exaggerated and should be rewritten. The 2 time points are actually limitations of the study.
Line 255 Citation needed
Methods
Lines 297-298 Ona and Inco are all one word - eg onabotulinumtoxinA
There is no inclusion criterion (5) indicated
Line 305 The number screened should be clear, not "at least 504"
Author Response
This is a well-written manuscript that is clear and which provides new data on the use of BoNT for spasticity treatment in a carefully selected group of patients.
REPLY: Thank you for your support and positive comments
My minor comments are as follows.
The abbreviation "mons" should be changed to months.
REPLY: use find and replace function to make such change throughout the article
Figures 4 & 5 are not correctly mentioned in the main text.
REPLY: Thanks for careful reading. They have been corrected.
Lines 180-181 & 286-287 These state about a return to baseline levels at 3 months, including MAS. But lines 111-112 state that MAS scores did not return to baseline after 3 months.
REPLY: again, thanks for careful reading. To be more accurate, the following changes have been made:
Lines 111-112: did not fully return….
Lines 180-181 & 286-287: return towards the baseline…
Lines 209-211 Not required - repeats earlier
REPLY: We prefer to keep this way. It makes transition and discussion (flow) easier.
Line 215 Citation 44 relates to a rat study, not a human study. So this statement about translation to humans needs detailed justification. How are such animal studies translational to humans? Evidence indicates much higher doses of BoNT are need for animal effects than for humans.
REPLY: Maybe the word “translational” is confusing here. We have revised the transition sentence to: Similar findings were observed in human subjects” (line 217)
Lines 215-218 Citation needed
REPLY: the same reference (#45), which covers lines 218-223
Lines 250-254 The present study only examined 2 time points after BoNT injections. Therefore these statements are a little exaggerated and should be rewritten. The 2 time points are actually limitations of the study.
REPLY: We agree a longer follow up duration will be better, thus we have added this limitation (lines 285-286) like this “However, a longer duration of follow up is needed to better reflect changes after BoNT injections.”
On the other hand, since BoNT injections usually are repeated every 3~4 months, our assessments do provide evidence to support the established clinical profile of BoNT effects. To be more specific, we have added “…. In an injection cycle” (line 256)
Line 255 Citation needed
REPLY: a reference is cited
Methods
Lines 297-298 Ona and Inco are all one word - eg onabotulinumtoxinA
There is no inclusion criterion (5) indicated
Line 305 The number screened should be clear, not "at least 504"
REPLY: thank you for careful reading: these changes are made: Toxins are single words; criterion 6 should be 5; and deleted “at least”
This manuscript is a resubmission of an earlier submission. The following is a list of the peer review reports and author responses from that submission.
Round 1
Reviewer 1 Report
Thank you for giving me the opportunity to review this manuscript. The topic is of interest to clinicians. Analyze biomechanical, electrophysiological and neuro-motor effects after botulinum toxin injections in elbow flexors among patients with chronic stroke survivors with spastic hemiplegia. Nevertheless, after a careful review, the paper has some methodological flaws.
Title
No Identify the type of study in the title.
Introduction
This study does not justify correctly the hypothesis and objectives.
Methods
Study design does not appear.
No Describes the scope of study, relevant places and dates, including periods of recruitment, treatment, monitoring and data collection.
It does not provide the sources and methods of selection of participants.
It does not describe the Origin (institutions) in which the data were recorded.
No indicates what measures have been taken to address possible sources of bias.
It does not explain how the sample size was determined.
It is a small sample size, only twelve stroke subjects with spastic hemiplegia, and different gender.
Therefore, the results cannot be extrapolated.
Results
It remains to describe and indicate the number of participants in each phase of the study; e.g. number of eligible participants, screened for inclusion, confirmed eligible, including in the study, those who had a complete follow-up and those who were analyzed. The reasons for the loss of participants in each phase are not described either. Clinical results are not maintained over time. Clinical results are not maintained over time, therefore it would be necessary to try other doses and more samples.
Discussion
It does not explain study limitations. The limitations of the study should have been discussed, taking into account possible sources of bias or of imprecision. Reasoning about both the direction and the magnitude of any possible bias. The limitations of the study should have been discussed, taking into account possible sources of bias or of imprecision. Reasoning about both the direction and the magnitude of any possible bias.
The generalizability of the results is not discussed (external validity).
Conclusion
The conclusions are not decisive, because being few subjects, it is not a conclusive study.
Reviewer 2 Report
This study investigates the effects of Botulinum neurotoxin injection on muscle resistance in stroke subjects. 12 stroke subjects received 100 units of BoNT to the spastic biceps brachii muscles, then the clinical effects are assessed by several methods at one week before (pre-injection), 3 weeks after, and 3 months after BoNT injection. The authors found that BoNT injections significantly reduced elbow flexor spasticity, muscle strength, reflex torque, and CMAP amplitude of biceps brachi at week 3, but not at month 3 among those subjects.
The manuscript is generally well written; however, data presentations are not sufficient to reach these conclusions, shortcomings are listed below.
Introduction:
Line35-44, The authors raised problems regarding hyperexcitability of spinal motor neurons with ref.no10-12, however, the study did not address this issue. Thus, I’m not sure if these sentences here are required or not.
Method:
- Ethical concern, there is no mention of informed consent of the subjects.
- There is no information on how 12 study participants were recruited from many patients with post-stroke. Selection bias needs to be considered.
- There is no information on how many times each subject has received BoNT injections in the past. This information is very important because the results of the study may differ by whether the subject was a responder or naïve to the BoNT injection.
- Evaluation of spasticity using MAS requires interpretation regarding the validity of the results, as it depends on the subjectivity of the evaluator and the relationship between the examinee and the evaluator. In particular, it may be difficult to clearly define the difference between 1, 1+, and 2. This assessment bias needs to be considered.
- Line 355, CMAP: There is no description of the circumstance for CMAP recordings, amplitudes of CMAP depend on not only the stimulus intensity, but also recoding circumstances (temperature, etc..), and peripheral nerve conditions. In addition, I questioned why the authors did not measure the area of CMAP. In CMAP recordings, it is often experienced that the shape of CMAP wave differs by the site of the recording electrode even in the same person. To study physiological changes of CMAP, at least three factors: Amplitude, Area, and Duration, should be reported simultaneously [PMID:16876477]. Did the authors confirm that a shape of a CMAP wave was consistent within the same individual?
- I questioned why the authors did not take f-wave recordings to see neuronal change at the same time as CMAP recordings, although the authors stated sentences regarding hyperexcitability of spinal motor neurons with ref.no10-12.
- Line341, Neuro-motor performance: CV, How many times elbow flexor isometric force was measured for each individual/each measure?
- In the fig4, SEMs appear to be big. I’m not sure whether One-way repeated ANOVA is an appropriate method or not. A GLM repeated measure would be better.
- What kinds of post-hoc test was used for each analysis?
- I could not find how the MAS score of 1+ was quantified for inclusion in the statistics.
Result:
Presentations of results are hard to follow. There is neither figure nor table regarding “CV results”.
Discussion:
Line177 [3.1. BoNT Effects], I do not agree with a discussion about “neural component”, because the authors did not address the assessment of “pure neural component” here.
Reviewer 3 Report
the manuscript is well written and comprehensive. it is worthy of publication in the present status.